# Proteostasis Disturbances and Inflammation in Neurodegenerative Diseases

**DOI:** 10.3390/cells9102183

**Published:** 2020-09-28

**Authors:** Tuuli-Maria Sonninen, Gundars Goldsteins, Nihay Laham-Karam, Jari Koistinaho, Šárka Lehtonen

**Affiliations:** 1A.I. Virtanen Institute for Molecular Sciences, University of Eastern Finland, Kuopio, Neulaniementie 2, 70211 Kuopio, Finland; tuuli-maria.sonninen@uef.fi (T.-M.S.); gundars.goldsteins@uef.fi (G.G.); nihay.laham-karam@uef.fi (N.L.-K.); jari.koistinaho@helsinki.fi (J.K.); 2Neuroscience Center, University of Helsinki, Haartmaninkatu 8, 00014 Helsinki, Finland

**Keywords:** neuroinflammation, immunoproteosome, ER stress, protein misfolding, ROS, pro-inflammatory cytokines, lipid peroxidation, advanced glycation end-products

## Abstract

Protein homeostasis (proteostasis) disturbances and inflammation are evident in normal aging and some age-related neurodegenerative diseases. While the proteostasis network maintains the integrity of intracellular and extracellular functional proteins, inflammation is a biological response to harmful stimuli. Cellular stress conditions can cause protein damage, thus exacerbating protein misfolding and leading to an eventual overload of the degradation system. The regulation of proteostasis network is particularly important in postmitotic neurons due to their limited regenerative capacity. Therefore, maintaining balanced protein synthesis, handling unfolding, refolding, and degrading misfolded proteins are essential to preserve all cellular functions in the central nervous sysytem. Failing proteostasis may trigger inflammatory responses in glial cells, and the consequent release of inflammatory mediators may lead to disturbances in proteostasis. Here, we review the mechanisms of proteostasis and inflammatory response, emphasizing their role in the pathological hallmarks of neurodegenerative diseases such as Alzheimer’s disease, Parkinson’s disease, and amyotrophic lateral sclerosis. Furthermore, we discuss the interplay between proteostatic stress and excessive immune response that activates inflammation and leads to dysfunctional proteostasis.

## 1. Inflammation Produces Proteostatic Dysfunction 

Inflammation and the disruption of proteostasis manifest upon normal aging and in some age-related neurodegenerative diseases. The neuroinflammation is considered as a beneficial physiological response within the brain or spinal cord, promoting the clearance of neuronal debris and assisting in tissue repair. However, uncontrolled and sustained inflammatory signaling can contribute to a variety of chronic inflammatory diseases. In the central nervous system (CNS), brain resident microglia and astrocytes are the primary sources of inflammation. Under pathological conditions, these glial cells facilitate the events that promote a neurotoxic environment [1,2]. Since neurons are primarily non-dividing cells and have a limited regenerative capacity, excessive neuronal death in the CNS has consequences on the motor, cognitive, and memory functions that are typically seen in patients of Parkinson’s disease (PD) and Alzheimer’s disease (AD), respectively. Therefore, inflammation has been considered as a contributor to neurodegeneration, together with glial activation and peripheral immune infiltration [3,4]. 

Proteostasis i.e., protein homeostasis, is the process of maintaining the intracellular and extracellular functional proteins. Proteostasis ensures the proper folding of newly synthesized proteins by mechanisms including the regulation of protein translation as well as the unfolding, refolding, and degradation of misfolded proteins. This process is essential, as 33–35% of newly synthesized proteins are prone to misfolding [5]. The control of proteostasis in postmitotic neurons becomes especially important with aging when the selection of proteostasis machineries are reduced, causing an accumulation of damaged proteins and organelles [6]. The two main protein degradation systems are the ubiquitin–proteasome system (UPS), which is responsible for the degradation of both functional and dysfunctional short-lived proteins coupled with ubiquitin molecules [7,8], and the autophagy-lysosomal system that degrades long-lived proteins, large aggregates of proteins, cellular components and organelles through the lysosomal compartment [9].

In this part of the review, we describe how inflammation causes proteostasis disturbances through the induction of reactive oxygen species (ROS) or reactive nitrogen species (RNS), leading first to the oxidative modification of proteins followed by protein misfolding. We also describe how subsequent dysregulation in the endoplasmic reticulum (ER), UPS, and autophagy leads to proteostatic dysfunction in neurodegenerative diseases (Figure 1).

### 1.1. Inflammation Induces Oxidative Stress

Inflammation is a protective response of a multicellular organism to injury. The function of inflammation is to localize, eliminate, and remove harmful stimuli and to recover damaged tissues. There has been evidence that ROS are involved in the initiation and progression of the inflammatory response [10]. ROS have important physiological functions such as the oxidation of cysteines, which is a necessary step in forming disulfide bonds into proteins [11]. However, excessive ROS production can cause oxidative stress, which is defined as disequilibrium between ROS production and the ability to detoxify the reactive oxygen intermediates. The extreme production and release of ROS have been proposed as a general pathological mechanism in all major chronic neurodegenerative diseases, including AD, PD, amyotrophic lateral sclerosis (ALS), Hungtington’s disease (HD), and multiple sclerosis (MS). Since oxidative stress can induce cell death and promote inflammation [12], cells have a battery of antioxidizing molecules and enzymes to prevent ROS accumulation. In a healthy state, mediators of oxidative stress and inflammation are in balance with the counteracting detoxifying and anti-inflammatory molecules. This balance is disturbed in some pathological states, and it is shifted toward the oxidative stress and pro-inflammatory direction, leading to DNA and protein damage, inflammation, and neuronal cell death. The accumulation of ROS under oxidative stress conditions results in the induction of protein oxidative modifications, including lipid peroxidation and glycoxidation reactions.

### 1.2. ROS and RNS Cause Protein Oxidative Modification Leading to Protein Misfolding

#### 1.2.1. Lipid Peroxidation

ROS can cause protein oxidative modification and lipid peroxidation at the cellular level, resulting in the generation of 4-hydroxy-2-nenotal (HNE). Due to its high reactivity, HNE forms protein adducts that cause protein misfolding and disturbances in the protein function. Moreover, HNE can induce carbonyl stress and deplete the antioxidant capacity of the cells. Evidence for lipid peroxidation has been found in AD, PD, ALS, and HD. In AD, elevated levels of HNE have been confirmed in patients [13] and found to target multiple proteins and enzymes. HNE affects the enzymes involved in the elimination of amyloid β-protein (Aβ), which are key enzymes of energy metabolism including aldolase, enolase, aconitase, and ATP synthase [14], as well as enzymes involved in antioxidant defense, such as superoxide dismutase, heme oxygenase, and peroxiredoxins [15]. In PD, the formation of HNE-alpha-synuclein adduct increases the oligomerization potential, thus triggering alpha-synuclein (α-SYN) aggregation [16]. Also, Lewy bodies (LB) stain positively for HNE in PD brains [17]. Besides the direct effect with α-SYN, HNE binds to the dopamine transporter and inhibits dopamine uptake, enhancing the progression of PD [18]. HNE levels are elevated in the cerebrospinal fluid of patients with ALS [19]. HNE was also found to colocalize with huntingtin inclusions in the striatal neurons, and HNE adducts are present in the caudate and putamen of HD brains [20]. 

#### 1.2.2. Advanced Glycation End-Products

The elevated ROS production can also lead to the formation of advanced glycation end-products (AGEs). Protein glycation is a process in which monosaccharides modify free amino groups of proteins. During this reaction, various intermediate compounds and eventually AGEs are formed. In AD, glycation plays a key role in the formation of amyloid protein, and high levels of AGEs have been observed in fractions of brain plaques. Furthermore, immunohistochemical stainings have demonstrated the presence of AGEs in neurofibrillary tangles and senile plaques [21]. In PD, the glycation of α-SYN is one of the important factors leading to aggregation and LB formation [22]. AGEs are colocalized with α-SYN and accelerate the aggregation process [23]. Glycation was also detected in the spinal cord and brain of ALS patients, and further related to ALS, increased levels of AGEs have been found in the presence of the copper–zinc superoxide dismutase (Cu, Zn-SOD-1) mutation that causes ALS [24]. These results suggest that glycation is responsible for the oxidative stress that culminates in neurodegenerative diseases. 

#### 1.2.3. Reactive Nitrogen Species 

In addition to ROS, reactive nitrogen species (RNS) are able to contribute to oxidative stress. RNS is derived from nitric oxide (•NO) and superoxide (O_2_•^−^) produced via the enzymatic activity of inducible nitric oxide synthase 2 (NOS2) and NADPH oxidase. RNS acts together with ROS to damage the cells and cause nitrosative stress. Among the RNS, especially the highly reactive peroxynitrite, (ONOO^−^) is known to induce lipid peroxidation and cause DNA damage [25]. RNS generation also modifies cysteine residues in proteins through S-nitrosylation or nitrotyrosination. The latter has been described in several neurodegenerative diseases linked to oxidative stress. NO production has been directly associated with neuroinflammation, especially with the inflammatory glial response (either astrocyte or microglia) [26]. NO-induced glial activation has a detrimental effect on neurons in AD, PD, and MS [27]. Susceptibility to NO and ONOO^−^ depends on the intracellular antioxidants and stress resistance signaling pathways. High levels of NO metabolites were also detected in post-mortem brains from patients with ALS along with protein damage caused by oxidation [28,29].

### 1.3. Misfolded Proteins Promote ER Stress 

One of the typical pathological hallmarks of many neurodegenerative diseases is the accumulation of misfolded proteins within the ER of neurons and glia. ER serves many functions, including folding and correcting the folding of newly synthesized proteins, the disposal of misfolded proteins, and trafficking proteins to the Golgi apparatus. The disturbance and imbalance between the load on the ER functions and its capacity lead to ER stress. ER stress triggers the unfolded protein response (UPR) in the ER in order to return the ER to its normal physiological balance [30]. The activation of UPR turns on a mechanism that allows cells to deal with the accumulated unfolded proteins [31]. While moderate stress enhances cellular protection by altering the transcriptome and proteome of the cell, prolonged ER stress disrupts the protective mechanism of the UPR [32]. Then, the inability to restore ER functions induces cell death via apoptosis and exacerbates neuroinflammation.

Recent research indicates a profound interplay between the ER and oxidative stress, which is mediated by ROS and derived reactive carbonyls, converging at the redox imbalance between a reducing environment in the cytosol and an oxidative ER, respectively [33,34]. ER stress may be both a trigger and a consequence of chronic inflammation. Chronic inflammation is often associated with diseases that arise because of primary misfolding mutations and ER stress. Similarly, ER stress and activation of the UPR are features of many chronic inflammatory and autoimmune diseases [35,36,37]. Next, we describe how dysregulation in the ER, UPS, and autophagy leads to proteostatic deficit in neurodegenerative diseases. 

### 1.4. Dysfunction of Cellular Proteostasis in Neurodegenerative Diseases

ER is a key contributor to proteostasis and UPR controls proteostasis. The UPR and ER-associated degradation (ERAD) interacts in a coordinated manner with the UPS and autophagy–lysosomal system to alleviate protein misfolding or its consequences. It is generally accepted that proteostasis deficits are linked to various neurodegenerative diseases, including AD, PD, and ALS disorders that are characterized by neuronal loss in different regions of the CNS. Even though all these diseases have different clinical outcomes, they all feature the accumulation of protease-resistant misfolded and aggregated pathological proteins. 

#### 1.4.1. Proteostasis in Alzheimer’s Disease

AD is the most common cause of dementia characterized by progressive cognitive and memory decline. The neuropathology includes the extracellular deposition of Aβ in the hippocampus and cortex as well as the formation of intracellular neurofibrillary tangles consisting of hyperphosphorylated tau protein. The Aβ peptides are derived from the amyloid precursor protein (APP) cleaved by beta-secretase and gamma-secretase to yield Aβ. Mutations in *APP* and *PSEN 1* and *2* genes account for about 5% of all AD cases, while the remaining cases are sporadic. For these patients, the risk is determined by a combination of genetic and environmental risk factors as well as aging.

Typically, the production of Aβ is counterbalanced by its elimination via processes including proteolytic degradation, cell-mediated clearance, or clearance from the brain into the peripheral blood circulation through passive and active transport. Several enzymes including neprilysin, insulin-degrading enzyme (IDE), endothelin-converting enzyme (ECE), and angiotensin-converting enzyme (ACE) have been reported to be capable of degrading Aβ [38,39]. Additionally, the UPS serves as a major regulator of Aβ accumulation in neuronal cells, either by decreasing the production of Aβ or promoting its proteolytic degradation [40]. Therefore, dysregulation in the UPS and/or an inability to clear Aβ deposits completely leads to Aβ accumulation in neurons’ cytoplasm, facilitating Aβ plaques formation. Furthermore, the ER protein membralin, which is an essential component of the ERAD complex mediating the degradation of ER luminal and membrane substrates, was shown to be downregulated in AD, suggesting a critical role for ERAD in AD pathogenesis [41]. Moreover, UPR activation has been demonstrated to correlate with the neuropathology (Braak stages) of AD and with the phosphorylation of ER stress transducer inositol-requiring enzyme-1α (IRE1α) [42]. IRE1 controls the expression of transcription factor XBP1. Interestingly, the polymorphism of the XBP1 promoter was suggested to be a risk factor to the development of AD. Furthermore, *PSEN1* inhibits IRE1α function [43,44]. Tau, another characteristic protein associated with AD, is an ultrastructural protein that can be degraded by both autophagy and UPS based on its conformation; for example, hypoacetylated tau is preferentially degraded by UPS. On the other hand, soluble and phosphorylated tau can interact with ERAD components and result in the activation of the UPR [45]. The accumulated tau impedes the clearance of ubiquitinated proteins from the ER and causes an ER stress response [45]. 

#### 1.4.2. Proteostasis in Parkinson’s Disease

PD is the second most prevalent neurodegenerative disease that mainly affects the motor system. PD is characterized by a gradual loss of dopaminergic (DA) neurons in the substantia nigra par compacta and the presence of inclusions known as LB and Lewy neurites. Alpha-SYN fibrils are the main component found in these inclusions located either in neuronal cell bodies or neuronal dendrites and axons. The progressive accumulation of α-SYN can be linked to the disruption of the UPS [46] and different types of autophagy [47,48]. Pathologically, α-SYN can affect the functions of several organelles, including ER, Golgi, proteasomes, lysosomes, and mitochondria. The aggregated form of α-SYN can bind to lysosomal membrane proteins and block their function [47]. It can also inhibit certain enzymatic activity domains of proteasomes [46] and the expression of proteins relevant to autophagosome assembly [48]. It leads to the inefficient removal of aggregated proteins due to the impairment in macroautophagy. Mutant α-SYN accumulates in the ER, where it can impair protein trafficking from ER to Golgi by interaction with Ras-related protein Rab-1A [49]. 

While both UPS and autophagy can clear α-SYN, the main pathway for its degradation appears to be lysosomal [50,51]. α-SYN can be degraded by macroautophagy and chaperone-mediated autophagy (CMA) depending on the structure of the aggregate and possible mutations in genes associated with PD [52]. Small soluble forms of α-SYN are degraded by CMA. Still, in the pathological condition, the burden shifts to macroautophagy. Yet, both pathways can compensate for each other. Misfolded α-SYN undergoes alternatively refolding in the ER. However, excessive refolding upregulates protein disulfide isomerase (PDI) reduction. PDI is a chaperone that assists oxidative refolding by forming disulfide bonds in proteins [53]. The re-oxidation of PDI is linked to an increase of hydrogen peroxide (H_2_O_2_), causing the release of cytoplasmic calcium from the ER through the dysregulation of inositol trisphosphate receptor. Released calcium may activate calpain and eventually lead to apoptosis. We have previously shown that PDI’s pharmacological inhibition by bacitracin prevents ER redox imbalance and downstream pro-apoptotic events [34]. Importantly, with age, lysosomal functionality is found to be dramatically impaired, and this could be one of the contributing factors to α-SYN pathology.

Several genes relevant for the onset of PD are involved in or interact with the autophagy–lysosomal system, including mutations in the GBA1 and ATP13A2 (PARK9) genes [54]. *GBA1* encodes the lysosomal hydrolase Gcase and PARK9 encodes the lysosomal ATPase. When these genes are mutated, they impair lysosomal activity and disrupt the autophagy process. In addition to the mutations in genes coding for lysosomal components, other PD-associated mutations have been implicated in the process of autophagy. Among those are mutations in the gene encoding vacuolar protein sorting-associated protein 35 (VPS35) responsible for endosomal–lysosomal trafficking and mutations in Parkin (PARK2), PINK1 (PARK6), DJ-1 (PARK7), and Fbxo7 (PARK15), which have been linked to the process of mitophagy involving the degradation of dysfunctional mitochondria by autophagy [54,55]. PINK1 interacts with Parkin and promotes the selective autophagy of damaged mitochondria [55]. Moreover, mutated LRRK2 (PARK8) impairs CMA, leading to the accumulation of α-SYN [56] as well an increased phosphorylation of leucyl-tRNA synthetase impairing autophagy [57]. 

#### 1.4.3. Proteostasis in Amyotrophic Lateral Sclerosis

ALS is characterized by the progressive damage of motor neurons, causing loss of muscle control. Pathologically, the nuclear TAR DNA-binding protein 43 (TDP-43) was identified as a key component of the insoluble and ubiquitinated inclusions in ALS patients’ brains [58,59]. TDP-43 protein deposition in cytoplasm occurs concomitantly with the depletion of its native form from the nucleus [58]. The cytosolic aggregates are known to be toxic. In addition, they can recruit nuclear TDP-43 and thus contribute to nuclear loss-of-function. As a consequence of the combination of the nuclear loss-of-function and cytosolic gain-of-function of TDP-43, motor neurons gradually degenerate in the brain and the spinal cord of patients with ALS. The accumulation of cytosolic TDP-43 is turned over mainly by UPS even though both degradation pathways—UPS and the autophagy–lysosomal system—are active. The crosstalk exists between the two clearance systems [60], and the inhibition of one clearance pathway renders the remaining one more effective [61]. Furthermore, many mutations associated with ALS affect genes involved in UPS or autophagy-mediated degradation [40]. 

The ubiquitinated inclusions of TDP-43 are the major features of pathological TDP-43. The E3 ubiquitin ligase (Parkin) ubiquitinates TDP-43 via the ubiquitin lysines, K-48 and K-63 [62]. While K-48-linked polyubiquitin chains of TDP-43 are degraded by UPS, K-63-linked polyubiquitin chains of TDP-43 undergo autophagic removal. As suggested by recent data [63], autophagy can have a dual role in TDP-43-associated toxicity; it can either accelerate or slow down the disease progression. The vacuolar fusion machinery and the endo-lysosomal pathways are essential for the TDP-43 clearance and cell survival. Defective endocytosis caused by abnormal levels of TDP-43 has been detected in the frontal cortex tissue of an ALS patient [64]. Impaired endocytosis leads to an increase in TDP-43 aggregation, whereas enhancing endocytosis can reverse TDP-43 toxicity and spare motor neurons [64]. 

ALS patients with mutations in superoxide dismutase 1 (*SOD1*) or RNA-binding protein FUS (*FUS*) are negative for ubiquitinated inclusions of TDP-43 but immunoreactive for mutant aggregated Cu/Zn SOD1 and fused in sarcoma protein (FUS), respectively [65,66]. Likewise, TDP-43, mutant FUS demonstrates abnormal cytoplasmic redistribution and aggregation [65]. In C9orf72-related ALS, TDP-43 proteinopathy is present, but additional inclusions devoid of TDP-43 are p62/sequestosome-1 and ubiquitin-positive [67].

As already mentioned, PDI assists protein refolding in PD. In ALS, PDI has been reported to be upregulated in the spinal cords of sporadic ALS patients [68]. Furthermore, PDI co-localizes together with TDP-43 [69] and SOD1 [68]. PDI is usually seen as a beneficial molecule. Nevertheless, recent studies have demonstrated that misfolded protein accumulation increases PDI levels, promoting the cell death cascade [34,70]. Concomitantly with these findings, our lab has shown that UPR may lead to the activation of PDI-dependent NADPH oxidase (NOX) and thus contribute to neurotoxicity in ALS [71]. The accumulated mutant SOD1 can also impair the ERAD machinery by interacting with ERAD components and, therefore, induce ER stress by altering protein trafficking [72]. ER stress can also be induced by the interaction of ALS-linked vesicle-associated membrane protein (VAPB) variant with either IRE1α/XBP1 [73,74] or activating transcription factor 6 (ATF6) [75]. Both IRE1α/XBP1 and ATF6 are ER stress sensor proteins, and their activities are reduced upon the interaction of VAP protein with the ER stress signaling system. 

## 2. Immune Responses Induced by Dysfunctional Proteostasis in Neurodegenerative Diseases 

The cellular stress response is a major regulator of the proteostasis network in various scenarios of induced imbalance in proteostasis. A growing body of evidence indicates that immune reactions are induced by proteostatic stress, and excessive inflammation may contribute to dysfunctional proteostasis [76]. Here, we review the interplay between the immune response and proteostasis, particularly in the context of neurodegenerative diseases.

Tight crosstalk between ER stress and immune responses has been demonstrated in several studies. First of all, immunogenic lipids are produced upon ER stress in antigen-presenting cells (APC) that cause the activation of natural killer T-cells (NKT cells) [77]. The indispensable factors here are UPR mediators, IRE1α, and protein kinase R-like ER kinase (PERK). Secondly, ER stress may trigger an acute inflammatory response through regulated intramembrane proteolysis of ER membrane-anchored transcription factor cyclic adenosine monophosphate-responsive element-binding protein H (CREBH), which is required for the activation of acute-phase response genes [78]. Moreover, a key regulator of the inflammatory response, nuclear factor kappa-light-chain-enhancer of activated B cells (NF-kB), becomes activated upon ER stress through the interaction of IRE1α with TNF receptor asssociated factor 2 (TRAF2) [79,80]. In parallel, the ER stress-induced PERK-translation initiation factor 2α (eIF2α) signaling pathway suppresses protein synthesis, which results in an increased ratio of NF-κB to IκB (inhibitor of nuclear factor kappa B) and the promotion of NF-κB-dependent transcription [81]. On the other hand, dysfunctional proteostasis affects cells involved in both innate and adaptive immunity. The resulting misfolded protein accumulation may promote increased pro-inflammatory cytokine production [82] as well as contribute to the development of immune senescence [83]. 

### 2.1. Immune Response in Alzheimer’s Disease

In AD, the hallmark of dysfunctional extracellular proteostasis is Aβ deposition. The Aβ plaques cause microglial inflammatory activation, migration, and phagocytosis [84,85]. While the primary immune response results in the clearance of Aβ, sustained microglial activation produces reactive microgliosis, causing the exacerbation of AD pathology. This is associated with the decreased microglial capability to phagocytose Aβ and pro-inflammatory cytokine release [86]. Another hallmark of AD pathology, tauopathy, has been demonstrated to spread between the brain cells by direct secretion, ectosomal, and exosomal mechanisms [87]. In tauopathy, a key role is played by microglia, which may propagate tau-related pathology through exosome release [88]. 

Among multiple factors supporting microglial fitness to maintain extracellular proteostasis, a remarkable role is played by the triggering receptor expressed on myeloid cell 2 (TREM2). TREM2 signaling is essential for microglial activation and survival, particularly for Aβ deposition determined responses [89]. Recently, a link between TREM2 signaling and microglia metabolic activity relying on mammalian target of rapamycin (mTOR) signaling was described, indicating the importance of balanced energy metabolism in proper microglial function. 

### 2.2. Immune Response in Parkinson’s Disease

Alpha-SYN mediated neuroinflammation is evident in PD neurodegeneration [90]. The formation of α-SYN fibrils that is facilitated by mutations results in the deposition of protein inclusions and consecutive microglial activation [91]. Remarkably, the inoculation of preformed α-SYN fibrils leads to LB pathology due to spreading through intercellular transmission [92]. The fibrillar form α-SYN that is produced and released by neuronal cells binds Toll-like receptor 2 (TLR2) and activates microglial inflammatory responses [93], leading to neurotoxicity. Besides activated TLR2-induced inflammation, activation of the NLR family pyrin domain containing 3 (NLRP3) has been demonstrated to result in the formation of inflammasome [94]. Altogether, these findings suggest that non-cell-autonomous neurotoxic effects of α-SYN are mediated primarily by glial pro-inflammatory activation [95].

It has also been demonstrated that α-SYN-induced phagocytic activity releases pro-inflammatory cytokines and ROS in microglia by activating TLR4 receptor [96]. However, astrocytic α-SYN uptake was not dependent on TLR4 expression. Pro-inflammatory factors released from activated microglia induce also neuronal major histocompatibility complex class-I (MHC-I) expression, which may trigger the antigenic response and DA neuron death mediated by cytotoxic T cells [97].

### 2.3. Immune Response in Amyotrophic Lateral Sclerosis

In the mutant SOD1-expressing rodent models of ALS and in patients with sporadic ALS, UPR-related molecules such as stress sensor kinases, chaperones, and apoptotic mediators are induced at disease onset and end stage, which indicates the presence of disturbed proteostasis in the disease [71]. 

In an ALS rat model, SOD1 carrying G93A mutation is destabilized and aggregates, causing mitochondrial dysfunction and increased ROS production. The disulfide-reduced SOD1 is increased with the course of the disease. In contrast, the ER-resident chaperone PDI, which is capable of re-oxidizing disulfide bonds between cysteine residues of SOD1, increases upon progression of the disease [98].

Importantly, the early upregulation of PDI in the microglia of transgenic mutant SOD1 mice coincides with the expression of a UPR marker: growth arrest and DNA damage-inducible protein (GADD34) in the spinal cord glia [72]. The impact of the UPR-induced upregulation of PDI during the ER stress still remains a controversial matter, since, besides the adaptive function of increased protein refolding, PDI may also generate the production of ROS through NOX activation or contribute to hydroperoxide generation [34,99]. The ROS species released in turn provide support for the neurotoxic and inflammatory activation of microglia.

### 2.4. NLRP3 Inflammasome in Neurodegenerative Diseases

Inflammasomes are multiprotein complexes normally located in the CNS [100]. They are expressed in neurons, microglia, astrocytes, macrophages, and endothelial cells. Inflammasomes are part of the innate immune system and recognize pathogen-associated molecular patterns (PAMPs) and danger-associated molecular patterns (DAMPs) [101]. The assembly of the inflammasome results in the activation of caspase-1 and the subsequent release of pro-inflammatory cytokines IL-1β and IL-18 as well as the induction of pyroptosis [102]. The activation of inflammasomes has been reported in several neurodegenerative diseases, especially the nucleotide-binding oligomerization domain leucine-rich repeat and pyrin domaincontaining (NLRP) 3 inflammasome is now widely investigated. Activation of the NLRP3 inflammasome is a two-step process. The first step primes the inflammasome and requires activation of the the NF-κB pathway to upregulate the expression of NLRP3, caspase-1, and prointerleukin-1β (pro-IL-1β) through the stimulation of TLRs [103,104]. After priming, several stimuli such as ionic flux, extracellular ATP, ROS, and lysosomal rupture can activate the NLRP3 complex [105].

The role and activation of inflammasomes have been studied in several neurodegenerative diseases, including AD, PD, and ALS. Human and animal studies have shown that abnormal protein aggregation of Aβ, α-SYN, or SOD1 can activate microglia, induce IL-1β release, and activate the NLRP3 pathway (reviewed in [106,107]). In AD, Aβ can activate the NLRP3 in microglia to produce IL-1β through TLR4 [108]. Additionally, higher levels of NLRP3, caspase-1, IL-1β, and IL-18 were detected in peripheral blood mononuclear cells (PBMCs) from AD patients [109]. In AD transgenic mice, Aβ treatment induced a high level of caspase-1 and IL-1β in the brain tissue [110,111,112], whereas the inhibition of either NLRP3 or caspase-1 in an AD mouse model increased the clearance of Aβ by microglia, reduced the Aβ deposition, and improved cognitive impairment [112,113]. The dysfunction of the blood–brain barrier and the release of pro-inflammatory cytokines from endothelial cells are linked to AD. Recently, a study showed that Aβ can activate the NLRP3 inflammasome and the production of IL-1β and IL-18 in endothelial cells [114]. Furthermore, inhibition of the NLRP3 increased the endothelial properties and survival, suggesting a role of NLRP3 in blood–brain barrier dysfunction in AD.

NLRP3, caspase-1, and IL-1β were increased in PD patients’ PBMCs and plasma when compared to age-matched healthy controls [115,116,117]. In addition, increased levels of IL-1b and IL-18 levels have been detected in the cerebrospinal fluid of PD patients [118]. High mRNA and protein expression levels of NLRP3 inflammasome components were found in several PD animal models [119,120]. Several studies have now linked the α-SYN and NLRP3 inflammasome activation in PD. Increased plasma levels of α-SYN and IL-1β in PD patients have been shown to correlate with the motor severity in PD patients [115]. Moreover, the fibrillar form of α-SYN induced NLRP3–caspase-1 complex activation and the release of IL-1β in PBMCs, monocytes, microglia, and astrocytes [117,121]. Recently, the role of α-SYN and NLRP3 activation in astrocytes was demonstrated [121]. Mouse astrocytes treated with oligomerized α-SYN increased the expression levels of NLRP3, caspase-1, and IL-1β, indicating an important role for astrocytes in NLRP3-related neuroinflammation in PD.

The activation of the inflammasome and upregulation of NLRP3 and its components, caspase-1 and IL-1β, have been reported in ALS patients and ALS mouse models, suggesting a role of inflammasomes in ALS [122,123,124,125]. In the mouse SOD1G93A model, higher levels of caspase-1 and IL-1β in microglia contributed to the disease progression [126]. Additionally, lipopolysaccharide (LPS) can activate caspase-1 and lead to an increased release of IL-1β in SOD1G93A mice [126]. In addition to microglia, studies suggest a critical role of astrocyte NLRP3 inflammasomes in ALS. Increased levels of NLRP3, ASC, caspase-1, and IL-18 were found in post-mortem spinal cord tissue, and astrocytes were identified as the main NLRP3 inflammasome-expressing cell type [127].

## 3. Immunoproteasome and Neuroinflammation in Neurodegenerative Diseases 

### 3.1. Structure and Function of Immunoproteasome

Proper protein turnover, including protein translation and degradation, is crucial for cell signaling and especially for removing damaged, misfolded, or oxidized proteins. The autophagy–lysosomal pathway and UPS degrade proteins. The degradation of proteins through UPS is divided into two distinct steps. In the first step, the target protein is conjugated to multiple ubiquitin units by the coordinated activation of ubiquitin-activating (E1), conjugating (E2), and ligase (E3) enzymes, in an ATP-dependent manner. In the second part, the 26S proteasome complex recognizes the polyubiquitin chain and degrades the protein into peptides. The recognition is carried out by the regulatory 19S complex, which also aids the substrate entry to the proteolytic site by binding, deubiquitylating, and unfolding ubiquitylated proteins [128]. The catalytic 20S complex is hollow, cylindrical, and composed of two outer and inner rings [129]. The outer rings are composed of seven α-subunits that bind and aid the substrate translocation into the catalytic core. The proteolytic site is located in the inner rings. The standard proteasome has seven unique β-subunits, and three of these have catalytic activity. They include β1, β2, and β5 subunits with caspase-like, trypsin-like, or chymotrypsin-like activity, respectively. In addition to the S19 complex, other regulatory proteasome activators (PA) exist, including 11S (PA28), PI31, and PA200, which can alter and enhance the proteasome function.

Interferon γ (INFγ) or environmental factors, such as conditions that trigger oxidized stress, can induce an alternative assembly of the 20S proteasome, which is commonly named immunoproteasome (IP; Figure 2A) [130]. The standard subunits β1, β2, and β5 are replaced by the inducible subunits LMP2 (iβ1), MECL-1 (iβ2), and LMP7 (iβ5). The catalytic activities of MECL-1 and LMP7 are similar to β2 and β5 subunits in the standard proteasome. However, the LMP2 subunit exhibits a chymotrypsin-like activity instead of the caspase activity in the standard β1 subunit [131]. The standard proteasome is expressed constitutively in nearly all mammalian cells, whereas the IP expression is low at basal conditions, except in the immune cells. However, INFγ or environmental factors can drastically increase the assembly of IP also in nonimmune cells. Additionally, INFγ can induce PA28αβ expression, which increases the activities of the beta subunits.

The most studied function of the IP is related to the immune function and antigen presentation. IP is capable of generating antigen peptides, which are first complexed to MHC-I in the ER and then exposed on the plasma membrane to be presented to CD8+ T lymphocytes [132]. The changes in the β subunits in the IP lead to increased overall chymotrypsin activity, which aids the generation of antigen peptides with hydrophobic C-termini and improves their fit into the groove of MHC-I molecules [133,134,135]. This increases the repertoire of peptides generated for MHC presentation. The assembly of IP in response to INFγ is faster than in a standard proteasome, which helps expanding the peptide pool needed for efficient immune responses [136]. The halftime of IP is shorter, which prevents persistent immune activation. 

In addition to the role in immune functions, the IP has a role in responding to various stress factors. During oxidative stress, the IP is efficient, and in some cases, it is even better in selectively degrading oxidized proteins than the standard proteasome [137,138]. Both the selectivity and activity of the IP can be increased by binding with the 11S (PA28) regulator. Increased levels of H_2_O_2_ cause protein oxidation, while increased levels of H_2_O_2_ cause protein oxidation, but the IP and the 11S regulator together with the standard proteasome help maintain the homeostasis during H_2_O_2_-induced oxidative stress. Even low levels of H_2_O_2_ without any protein damage have been shown to increase the synthesis of IP, 11S, and standard proteasome [137,138,139,140,141]. This may help the cells to preadapt to a potential increase in oxidative stress and be more prepared to degrade higher levels of oxidized proteins. A low level of nitric oxide is an essential factor in regulating the vascular tone, but, again, high concentrations can cause oxidative damage [142]. High levels of nitric oxide upregulate IP and help cells cope with increased protein damage [143,144,145]. Cells that are naturally exposed to higher levels of nitric oxide, including endothelial cells, express correspondingly higher levels of IP. IP’s expression and activity are essential for cell survival in the environment with high NO levels.

### 3.2. Immunoproteasome Function in CNS

Previously, the CNS has been seen as an immune-privileged area because of the immunosuppressive environment and the absence of dendric cells. However, currently, the evidence supports the idea that the CNS is not isolated but is actively communicating with the immune system. Nowadays, neuroinflammation is seen as a complex interplay between the CNS and systemic cells. In the CNS, neurons and glia (microglia, astrocytes, and oligodendrocytes) continuously express low amounts of IP, which suggests that IP has a role in maintaining homeostasis in the CNS [146,147]. In addition, pro-inflammatory cytokines, including INFγ and TNFα, or oxidative stress generally induce the expression of the IP and the disassembly of the standard proteasome [148,149,150,151]. This is thought to enhance protein degradation and allow the cells to cope with the protein overload. The IP can degrade aggregation-prone proteins at the same or even higher rate and efficacy than the standard proteasome [152,153]. The expression of MHC-I in the CNS has functions beyond antigen presentation. MHC-I expression in neurons has been linked to early neuronal development, synaptic plasticity, axonal regeneration, memory, and reward [154,155,156,157]. Nevertheless, neurons and glia can act as professional APC and thereby prolong inflammation/oxidized stress. Consequently, the increased IP may make the CNS cells more vulnerable to auto-immune damage [147]. 

### 3.3. The Role of Immunoproteasome in Neurodegenerative Diseases

Several studies have shown increased IP activity in various neurodegenerative diseases, including AD, PD, ALS, HD, and MS (Figure 2B). The increased expression and activity of the IP might be beneficial in the early stages of neurodegeneration by compensating for the protein overload and decreased function of the standard proteasome. However, the persistent overactivity of the IP can enhance the neuroinflammation and lead to neuronal cell death.

#### 3.3.1. Immunoproteosome in Alzheimer Disease

Several human post-mortem studies and experimental models have shown changes in the IP function and activity in AD. Decreased gene expression of standard b5 subunit and increased gene and protein expressions of LMP7 (β5i) and MECL-1 subunits have been observed in the hippocampus of AD brains [158,159]. In addition, the activities of the IP subunits LMP7 (β5i), MECL-1 (β2i), and LMP2 (β1i) are increased in the hippocampus of AD brains, which also correlates with the tau pathology [158]. Overall, the expression of IP in AD patients has been reported to be elevated compared to non-demented elderly, while in the young brain, the expression may be barely detectable [160]. 

Increased IP expression has also been detected in several AD animal models. The IP activity was primarily raised in the AD mice cortex and amplified gene and protein expressions in neurons and glia surrounding the amyloid-beta plaques [158]. Decreased levels of standard β5 subunit and increased LMP2 (β1i) and MECL-1 (β2i) levels and trypsin activity were detected in AD mice [161]. The increased gene and protein expressions of LMP7 (β5i) and LMP2 (β1i) subunits also correlated with age and amyloid-beta pathology in AD mice [162].

#### 3.3.2. Immunoproteosome in Parkinson’s Disease

Increased levels and activity of LMP7 (β5i) subunits were found from post-mortem brains of PD and dementia with LB patients. Notably, the increase was detected in both neurons and glial cells in the substantia nigra area. In contrast, the increase was seen only in glial cells in the less vulnerable ventral tegmental area [163]. Increased activity of the LMP7 (β5i) subunit was also demonstrated in an experimental model using 6-hydroxydopamine (6-OHDA). The neurotoxin, 6-OHDA, upregulated the LMP7 subunit in DA neurons, both in vitro and in vivo studies [164].

#### 3.3.3. Immunoproteosome in Amyotrophic Lateral Sclerosis

Although evidence of the relation between the IP and ALS is missing from human studies, several experimental animal models of ALS have shown increased IP function. Proteasome activity was raised in the spinal cord of SOD1 G93A transgenic mice. The standard proteasome subunits 7 and 5 were expressed constitutively, but a marked increase in IP subunits LMP2 (β1i), MECL-1 (β2i), and LMP7 (β5i) were found. Additionally, the induction of IP subunits occurred mainly in microglia and astrocytes [165]. Other studies have also shown similar initiation of the IP in the spinal cord with SOD1 G93A transgenic mice, although the standard proteasome activity was found decreased. [166,167]. A study by Puttaparthi et al. showed increased proteasome activity and induction of the IP in the spinal cord of SOD1 G93A transgenic mice. Additionally, mice lacking the LMP2 (G93A SOD1/LMP2−/−) subunit did not exhibit a change in the motor function decline, suggesting that IP function does not alter the SOD1-induced behavioral phenotype [168].

#### 3.3.4. Immunoproteosome in Huntington Disease

Increased levels of LMP7 (β5i) and LMP2 (β1i) were detected in the cortex and the striatum of HD patients’ brain compared to age-matched controls [169]. The increase was related to decreased levels of the corresponding subunits of the standard proteasome. Moreover, the induction of the IP subunits in neurons was associated with neurodegeneration. Similarly, the IP subunits LMP7 (β5i) and LMP2 (β1i) were increased in neurons and glia in the striatum and cortex of HD mice [169]. 

#### 3.3.5. Immunoproteosome in Multiple Sclerosis

The IP and PA28ab regulator’s expression was detected in MS patients but was not seen in young controls. In addition, LMP2 (β1i) and PA28αβ were detected in the cortex and the white matter plaques in neurons and glia [170]. In the experimental model of MS, MOG-EAE mice (myelin oligodendrocyte glycoprotein experimental autoimmune encephalomyelitis), overall peptidase proteasome activity during the acute phase of EAE correlated with increased levels of LMP2 (β1i), MECL-1 (β2i), and LMP7 (β5i) subunits in neurons and glia. These findings were opposite in the chronic phase [171,172]. Similarly, the amount and activity of the IP subunits were increased in MOG-EAE rats [173]. In myelin Basic Peptide (MBP)-EAE mice, LMP2 (β1i) and LMP7 (β5i) subunits were increased, and LMP2 (β1i) was dominantly expressed in oligodendrocytes, whereas LMP7 (β5i) was mainly in brain-infiltrating lymphocytes [174]. 

### 3.4. Immunoproteasome Inhibitors in Neurodegenerative Diseases

The increasing evidence of the role of IP in inflammation and neurodegenerative diseases has raised the interest to target the IP for therapy. Several IP inhibitors have been developed for autoimmune diseases and some cancers [175]. Currently, KZR-616 inhibitor from Kezar Life Sciences is being tested in a clinical trial for Systemic Lupus Erythematosus with and without Nephritis (ClinicalTrials.gov Identifier: NCT03393013). While the IP inhibitors have beneficial effects in immune-related diseases, the preclinical results from neurodegenerative diseases have been contradictory. The impact of IP inhibition seems to be disease and context-dependent. While for AD and MS, the inhibition could be beneficial based on the current findings, for PD and ALS, the obtained results from animal models showed the opposite. 

In the AD mice models, IP inhibition improved mainly cognitive functions. Trasngenic APP-PS1 mice crossed with mice deficient for the IP subunit LMP7 (β5i) resulted in impaired IP function. These LMP7 (β5i)-deficient mice did not show significant Aβ pathology; however, microglia showed altered cytokine responses. The altered cytokine profile was associated with improved Aβ-associated cognitive deficits typically observed in APP-PS1 mice [165]. In another study, dual inhibition of IP subunits LMP2 (β1i) and cP catalytic subunit Y with YU102 ameliorated the cognitive effects in the AD mouse model [176]. The inhibition did not affect the Aβ deposition but suppressed the cytokine secretion from microglia cells.

In different experimental models of MS, proteasome inhibitors proved to be efficient to some extent [173,177]. The impact of IP inhibitor ONX 0914 was studied in two different mouse models of MS. ONX 0914 attenuated the disease progression in MOG_35–55_ and PLP_139–151_-induced-EAE [178]. The isolation of lymphocytes from the spinal cord revealed a substantial reduction of cytokine-producing CD4 cells in treated mice. These results suggest that IP inhibitors may have a potential for treating MS patients.

However, the fact that the IP inhibition could also have adverse effects was demonstrated in PD and ALS rodent models. While 6-OHDA upregulated the LMP7 (β5i) subunit in DA neurons, inhibition of the IP increased 6-OHDA-induced neurotoxicity both in vitro and in vivo [164]. A similar effect was seen in SH-SY5Y cells exposed to rotenone [179]. Knockdown of the ib1 subunit resulted in increased α-SYN accumulation, the degradation of tyrosine hydroxylase, the release of ROS, an increased level of malondialdehyde, and a decreased level of glutathione, and it also promoted apoptosis in SH-SY5Y cells after rotenone treatment. On the other hand, Oxyphylla A, the LMP7 (β5i) subunit activator, promoted α-SYN degradation in the cellular PD model [180]. Altogether, these results demonstrate the potential neuroprotective role of the IP in PD. 

Our laboratory studies have shown a potential neuroprotective effect of the IP in ALS when treating G93A-SOD1 transgenic mice with pyrrolidine dithiocarbamate (PDTC), which is an inhibitor of NF-κB [181]. The PDTC treatment completely blocked the IP expression and significantly decreased the survival of the mice. The exposure did not affect the standard proteasome. These results suggested that the IP may help the nervous system to cope with the harmful effects of SOD1-G93A mutation.

## 4. Conclusions

Cellular viability and functions are dependent on not only adequate protein production but also an efficient degradation of excess, damaged, and misfolded proteins. Thus, disturbances in cellular protein homeostasis or proteostasis can be detrimental to the cell. Here, we have reviewed the current research related to the relationship between proteostasis disturbances and inflammatory response in neurodegenerative diseases. A growing body of evidence shows that protein misfolding, aggregation, and aberrant modifications can lead to excessive immune responses causing neuroinflammation, which is associated with neurodegenerative diseases.

On the other hand, reactive glial cells in the CNS play an important role in deleterious non-cell-autonomous mechanisms leading to the loss of proteostasis. This vicious feed-forward loop may have a critical impact on neuronal viability. Particularly, failures in the clearance of aggregated proteins have been associated with aging and neurodegenerative diseases such as AD, PD, and ALS, especially in gene mutations connected with cellular proteostasis. Furthermore, excessive immune responses that initiate inflammation and lead to dysfunctional proteostasis are evident in AD, PD, and ALS. Further elucidation of the individual steps of proteostasis and inflammation, especially in human-based models, will provide a better understanding of the cellular processes and open a window/the way for the development of novel pharmacological strategies for neurodegenerative diseases. 

## Figures and Tables

**Figure 1 cells-09-02183-f001:**
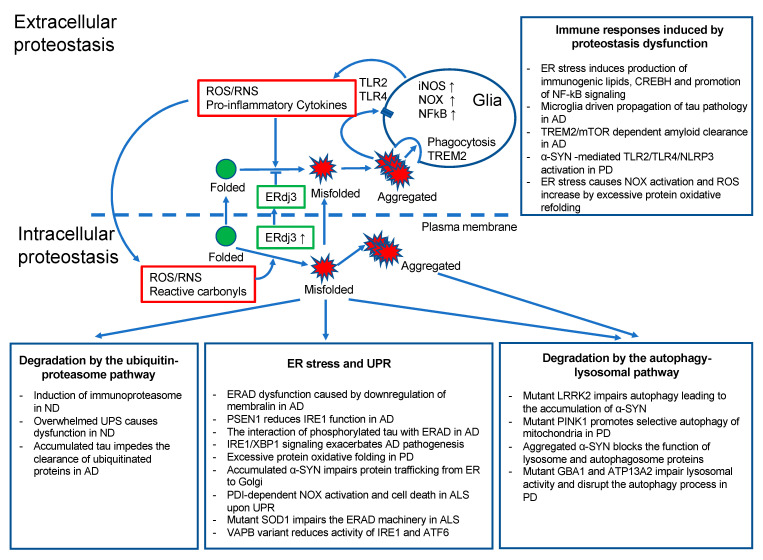
The extracellular and intracellular proteostasis in neurodegenerative diseases. Scheme representing proteostatic dysfunction in neurodegenerative diseases and the associated link between proteostasis and the inflammatory response. ND, neurodegenerative diseases; NF-kB, nuclear factor kappa-light-chain-enhancer of activated B cells; ERdj3, endoplasmic reticulum DnaJ homologue; TREM2, triggering receptor expressed on myeloid cells 2.

**Figure 2 cells-09-02183-f002:**
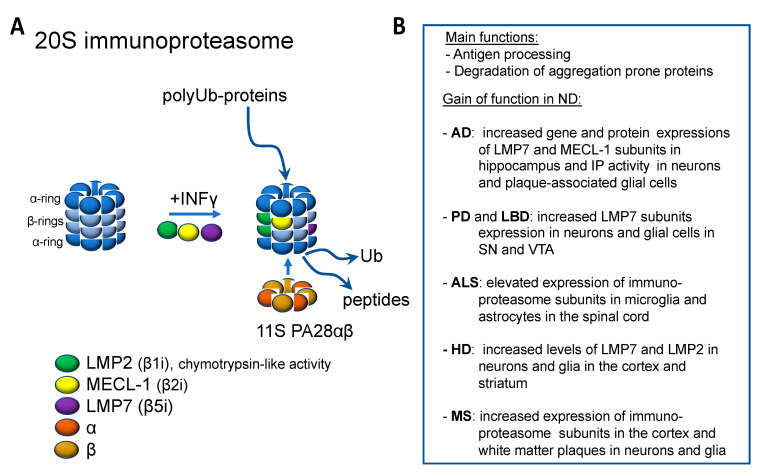
Immunoproteosome and its function in neurodegenerative diseases. (**A**) Immunoproteosome is formed from the constitutive proteosome upon inflammatory stimuli by the incorporation of specific β subunits and the co-production of proteosomal activator 11S PA28αβ. (**B**) Evidence for the increased function of immunoproteosome in neurodegenerative diseases.

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
