# Peer review of "Proteostasis Disturbances and Inflammation in Neurodegenerative Diseases"

_cells, 2020, doi:10.3390/cells9102183_

Round 1

Reviewer 1 Report

The manuscript by  Sonninen et al. entitled “Proteostasis disturbances and inflammation in neurodegenerative diseases” reviews the current state of research on the role of protein homeostasis and inflammation in neurodegenerative disorders such as AD, PD, HD and ALS. The manuscript covers various aspects of the above topic in a comprehensive and communicative way; 152 publications are cited in it.  It contains 2 figures supporting the conclusions made on the basis of cited articles. Thus, after a minor revision I highly recommend this manuscript to be accepted for publication.

Minor points:

Line 26, instead of “CNS” should be “central nervous system”

Line 70, instead of “lead” should be “leads”

Line 79, instead of “which a necessary” should be “which is a necessary”

Line 207, instead of “promotor” should be “promoter”

Line 305, comma should be added after “Here……”.

Abbreviations “IRE1”, UPS, ND should be developed.

Line 358, the term “MHCI” should be explained.

Author Response

OUR RESPONSE: We thank the reviewer for these valuable suggestions. All suggested changes by the reviewer have been corrected in the text and the abbreviations spelled out.

Reviewer 2 Report

The review entitled “Proteotoxicity and Neurodegenerative Disesases” Ruz and colleagues summarise the relationship between proteostasis disturbances and inflammatory response in AD, PD, ALS, and HD. This review article summarizes important issues focusing on the link between inflammation and proteotoxicity describing the specific mechanisms involved in analysed diseases.

The manuscript is well written, and it is easily readable.

Despite that contents are most adequately provided to support the authors’ viewpoints, there are few minor concerns that should be addressed.

1) The 2 figures need to be cited in the text.

2) In “proteostasis in ALS” part only TDP-43 protein has been described even if many other proteins that misfold and aggregate should be considered (at least SOD1, C9ORF72, and proteins involved in degradative pathways).

3) line 254 Fbxo should be changed with Fbxo7.

4) line 313 the complete name of CREBH should be spelled.

5) line 358 “MHCI” changed with “MHC-I”.

Author Response

1) The 2 figures need to be cited in the text.

OUR RESPONSE: Figure 2 has been cited twice: on line 401 as Figure 2A and on line 468 Figure 2B.

2) In “proteostasis in ALS” part only TDP-43 protein has been described even if many other proteins that misfold and aggregate should be considered (at least SOD1, C9ORF72, and proteins involved in degradative pathways).

OUR RESPONSE: We thank the reviewer for this valuable suggestion. The text on SOD1, C9orf72 and FUS was now included in a chapter entitled “Proteostasis in ALS”.

3) line 254 Fbxo should be changed with Fbxo7.

4) line 313 the complete name of CREBH should be spelled.

5) line 358 “MHCI” changed with “MHC-I”.

OUR RESPONSE: All suggested changes by the reviewer has been corrected in the text.

Reviewer 3 Report

This is a well-written review purporting to link proteostatic disruption with inflammation in neurodegenerative diseases.

Major weaknesses:

  1. Some logical misstatements in mechanisms. For example, the authors state that ROS-triggered HNE modification, AGE glycation, and other oxidative modifications of proteins can induce ER stress, although all of the ROS target proteins they cite are cytosolic, and not expected to exist in the ER. The link between oxidative stress and ER stress is somewhat obscure (Cao and Kauffman, "Endoplasmic Reticulum Stress and Oxidative Stress in Cell Fate Decision and Human Disease" Antioxidants and Redox Signalling, 2014). If the authors think that this link is secure, the appropriate references need to be cited.
  2. Although the immunoproteosome is extensively discussed in this review (which is primarily involved in microglial activation), there is no mention of inflammasome activation beyond synucleinopathies (e.g., NLRP3, which is expressed in neurons and glia), and is now thought to play a key role in neuroinflammation in all neurodegenerative diseases (including AD and ALS). The authors should expand on this.

Minor weaknesses:

  1. line 134: zink is zinc
  2. line 260, 361 494:Amyotrophic is not capitalized.

Author Response

Major weaknesses:

  1. Some logical misstatements in mechanisms. For example, the authors state that ROS-triggered HNE modification, AGE glycation, and other oxidative modifications of proteins can induce ER stress, although all of the ROS target proteins they cite are cytosolic, and not expected to exist in the ER. The link between oxidative stress and ER stress is somewhat obscure (Cao and Kauffman, "Endoplasmic Reticulum Stress and Oxidative Stress in Cell Fate Decision and Human Disease" Antioxidants and Redox Signalling, 2014). If the authors think that this link is secure, the appropriate references need to be cited.

OUR RESPONSE: We thank the reviewer for the valuable comments. Indeed, ER stress here was not very appropriate, so thanks to the reviewer for spotting this....we have now made the following corrections:  

  • In chapter 1: In this part of the review, we describe how inflammation causes proteostasis disturbances induces the endoplasmic reticulum (ER) stress through induction of reactive oxygen species (ROS) or reactive nitrogen species (RNS), leading first to protein oxidative modification followed by protein misfolding

  • In chapter 1.3 we added text with the reference suggested by the reviewer: Recent research indicates a profound interplay between the ER and oxidative stress, mediated by ROS and derived reactive carbonyls, converging at the redox imbalance between a reducing environment in the cytosol and an oxidative ER, respectively [33,34].

  1. Although the immunoproteosome is extensively discussed in this review (which is primarily involved in microglial activation), there is no mention of inflammasome activation beyond synucleinopathies (e.g., NLRP3, which is expressed in neurons and glia), and is now thought to play a key role in neuroinflammation in all neurodegenerative diseases (including AD and ALS). The authors should expand on this.

OUR RESPONSE: We thank the reviewer for this valuable suggestion. The chapter on NLRP3 inflammasome in neurodegenerative diseases was added.

Minor weaknesses:

  1. line 134: zink is zinc
  2. line 260, 361 494:Amyotrophic is not capitalized.

OUR RESPONSE: Both suggested changes by the reviewer has been corrected in the text.

Round 2

Reviewer 3 Report

Much improved. The section on the inflammasome is first-rate.